# Validity and reliability of the Polish version of the Self-Compassion Scale and its correlates

**Dagna Kocur** [1]☯*, **Maria Flakus**[1]☯, **Małgorzata Fopka-Kowalczyk**[2]☯

**1** Department of Psychology, University of Silesia in Katowice, Katowice, Poland, **2** Department of Pedagogical Sciences, Nicolaus Copernicus University in Toruń, Toruń, Poland

☯ These authors contributed equally to this work.
* dagna.kocur@us.edu.pl

## Abstract

This study adapts the Self-Compassion Scale into Polish and tests the validity, reliability and factor structure of its measures. In the first phase of the research (Study I), 645 respondents were assessed using the NEO-FFI Scale, the Self-Esteem Scale and a back-translated version of the Self-Compassion Scale. The aim of Study I is to analyse the factor structure of the Polish adaptation of the Self-Compassion Scale. The results of analyses using structural equation modelling and exploratory structural equation modelling confirm the six-component structure of the Self-Compassion Scale and the possibility of distinguishing a single primary factor. The results of these analyses indicate that self-compassion is conceptually distinctive from personality traits and self-judgement. In the second phase of the study (Study II), 688 respondents were assessed and the findings show that self-compassion is a predictor of depressive symptoms, trait anxiety, and satisfaction with life, and is also linked to emotional intelligence. In conclusion, the findings of this study show that the Polish version of the Self-Compassion Scale is a reliable and valid measure of self-compassion.

**Data Availability Statement:** All relevant data are within the OSF database (DOI: 10.17605/OSF.IO/38HVD; https://osf.io/38hvd/) and the Supplemental Information files.

## 1. Introduction

Over the last 20 years, we have seen a significant increase in the interest in self-compassion in the social sciences and the medical sciences. Although the concept of self-compassion has its roots primarily in Buddhism and Eastern philosophy, it was introduced into contemporary psychology by Kristin Neff in 2003 [1, 2], who defines it as self-understanding, self-kindness, and the ability to compassionately understand one's own suffering, limitations, failures, and difficult emotions, as well as a belief that everyone goes through difficult experiences. Self-compassion consists in experiencing a feeling of caring and kindness towards oneself and adopting an understanding and non-judgemental attitude towards one's own imperfections and failures [1]. Current international research and the few studies conducted in Poland, e.g., the 2011 study by Dzwonkowska [3] that use the Self-Compassion Scale (SCS) have demonstrated just how important self-compassion is for human functioning.

**Funding:** The author(s) received no specific funding for this work.

**Competing interests:** The authors have declared that no competing interests exist.

## 1.1. Tool design and factor analysis

According to the theory by Kristin Neff [1, 2], self-compassion has a three-component structure, or more specifically, six elements, as each positive element has a negative counterpart. The first component comprises *self-kindness*, the converse of which is *self-judgement*. This is understood as gentleness and understanding towards oneself, as opposed to sharp criticism. The second component comprises *common humanity*, the negative counterpart of which is *isolation*. This involves the individual's conviction that bad or difficult things happen not only to them but are characteristic of most people's experience. The third component comprises *mindfulness*, the negative counterpart of which is *overidentification*. It consists in continuing to be aware of one's experience, as opposed to exaggerating or ignoring specific aspects, such as the pain one is feeling [1]. Many replication studies have confirmed this six-factor structure [4–6], but confirmation has not been possible in all cases [7].

In studies on self-compassion, one often encounters assessment results presented using a broad scale with inverted negative subscales. However, results of highly detailed analyses are presented in a breakdown of six subscales: self-kindness, self-judgement, common humanity, isolation, mindfulness and overidentification. An equally frequent subdivision involves three factors: self-kindness, common humanity and mindfulness (in this case, the negative counterparts are also inverted). However, studies increasingly present the results of assessments in a breakdown comprising two factors, one positive and the other negative, i.e., self-compassion (with self-kindness, common humanity and mindfulness as subscales) and self-coldness (with self-judgement, isolation and overidentification as subscales).

Some studies even indicate that a separate analysis of the positive component (self-compassion) and the negative component (self-coldness) is crucial, with a heavy focus on self-coldness, which is also referred to as self-criticism [7–10].

The analyses conducted by Muris and Petrocchi [10] assess significant differences between the positive and negative components of self-compassion in the context of psychopathology. They conducted a meta-analysis of the findings of 18 studies on self-compassion and its variables related to psychopathology. The tests comparing the strength of the relationship between the positive and negative components of self-compassion and psychopathology show that the association between the negative indicators and mental health problems is stronger than that between the positive indicators and mental health problems. Other studies have also pointed to strong relationships between self-criticism (an element of self-coldness) and psychopathology [9]. According to Muris and Petrocchi [10], using the broad score of the SCS or the Self-Compassion Scale-Short Form (SCSSF) is likely to reflect an exaggerated relationship with psychopathological symptoms.

Research by Kane et al. [11] confirmed the significant differences between the positive and negative components of self-compassion for diabetes-related distress in type-2 diabetes patients. Research by López, Sanderman and Schroevers [12] also pointed to the significance of self-coldness, with findings indicating that there is a stronger relation between depressive symptoms and the negative SCS items and subscales than between depressive symptoms and the positive SCS items and subscales. Furthermore, Kane et al. found no significant evidence of the positive components of self-compassion playing a mitigating role in the relationship between self-coldness and depressive symptoms. Other studies have also pointed to well-being being more strongly related to self-compassion than to self-coldness, while distress has been found to be more strongly related to self-coldness than to self-compassion [13]. In addition, Pfattheicher et al. [14] challenge the use of the broad SCS score, arguing that self-coldness is identical to neuroticism—one of the *Big Five* in the five-factor model of personality.

However, critical opinions have also been expressed regarding the separate analysis of self-compassion and self-coldness. For example, in responding to a paper by Pfattheicher et al. [14] and using the same data but different methods, Neff, Tóth-Király and Colosimo [15] prove the legitimacy of the broad SCS score. Furthermore, they also demonstrated that self-compassion is strongly correlated with neuroticism and depression, which explains the significant incremental validity in *satisfaction with life* versus neuroticism, depression and anxiety. Neff et al. [16], in a study involving 20 diversified groups, highlighted the legitimacy of using a six-factor SCS structure or the SCS broad score, as opposed to analyses using the two-factor breakdown of compassionate and uncompassionate self-responding. The debate on the legitimacy of using the broad score, the six-factor breakdown, or the two-factor breakdown persists in the most recent studies in the literature [16–19].

However, the debate on the theoretical structure of the SCS is not necessarily limited to the simple question of whether self-compassion is a homogenous (i.e., comprised of only one factor) or heterogeneous (i.e., comprised of two or six factors) construct. Indeed, there is the possibility of accommodating both perspectives within a single latent trait theoretical model. In this context, contemporary studies discuss two possibilities: high-order and bifactor models. The use of both a broad score and subscales is justified in both models. Consequently, in both cases, the model permits the inclusion of both a higher-order factor (i.e., a general factor) representing a general construct and several first-order factors (i.e., specific factors) reflected in the subscale scores [20–23]. However, there is a dissimilarity, which lies in the relationships between a single item on the scale and a general construct. In the first option, the researcher assumes that the higher-order factor can only influence the responses on an individual scale item via first-order factors as a pathway. Thus, the relationship between a general factor and specific factors is hierarchical [20, 21]. However, it is also assumed that the general factor and the specific factors coexist in the latter option. Consequently, there is a direct association between the general factor and the responses to individual items. Hence, within a bifactor model, a group of specific factors do not correlate [22, 23].

Previous research studies have found little support for higher-order models, either regarding a justification for using both a single broad score and six subscale scores [2, 24–26] or for using two broad scores and six subscale scores. However, in the case of the latter, the research is primarily exploratory and no confirmatory analytical approach was applied to confirm two-factor models [7]. However, it has been argued that a bifactor model can provide an excellent theoretical explanation of self-compassion, assessing behaviours that are directly representative of both self-compassion and its specific components [27]. Indeed, some studies have proved that the six-factor correlated model and the bifactor model tend to have an acceptable fit in at least some of the assessed subsamples, e.g., in the USA (undergraduate, community and meditator samples) [28], and in the general population of the following countries: Scotland, Franc, Brazil, Portugal and Italy [29–32]. Notably, the studies mentioned earlier reveal that between 90% and 94% of the variance in scores is attributable to a general self-compassion factor. This can be interpreted as major support for the existence of a general self-compassion factor, alongside six specific subcomponents. Consequently, various two-factor solutions were consistently rejected in these studies, indicating an inadequate fit [28, 29].

However, the dimensionality of self-compassion is still in question. For example, no support was found for a single-bifactor model in research conducted by Montero-Marín et al. [33], who assessed samples of Spanish and Brazilian-Portuguese doctors. Interestingly, they found support for two higher-order factors and six subfactors. Similarly, Brenner et al. [34] argued that a two-bifactor model with six specific factors tends to achieve a more acceptable fit than a single-bifactor model with a sample of U.S. undergraduates. Against this backdrop, a multigroup analyses conducted by Tóth-Király and Neff [35], involving 18 samples and 12

different languages (N = 10,997), provides evidence of the configural, weak, strong, strict and latent variance–covariance of the single-bifactor exploratory structural equation modelling (ESEM) model of the SCS across different groups. These findings indicate that the SCS provides an assessment of self-compassion that is psychometrically equivalent across groups. Nevertheless, the findings of a study comparing latent mean invariance across populations indicate that levels of self-compassion differ across groups. The study shows that global levels of self-compassion are lowest in clinical populations and highest in student populations. Another argument supporting a non-divisional approach to self-compassion and self-coldness is provided by the outcomes experienced by individuals who attended self-compassion development workshops, among whom a post-study detected an even increase in scores on all five subscales (except the mindfulness subscale, on which the increase was small) compared to a pre-study [27].

## 1.2. Testing the theoretical structure of self-compassion using exploratory structural equation modelling (ESEM)

An exciting aspect of the debate on the theoretical structure of self-compassion is the proposed methods for validating this structure. Although confirmatory factor analysis (CFA) appears to be an appropriate tool for testing the adequacy and fit of a model—and is therefore very often used to confirm the theoretical validity of psychometrical measures—recent studies in the literature indicate that ESEM may be a promising alternative to CFA approaches, primarily because of the model specification procedure and the way non-target loadings within ESEM models are treated. Proper use of CFA obliges the researcher to specify the a priori parameters of the proposed model in terms of the associations between observed and latent constructs. Typically, this approach results in a situation in which each indicator of the latent variable is linked to only one factor, while other possible loadings are constrained to zero [36]. In contrast, ESEM requires only information on the number of latent factors, with a free estimation of the other parameters. In ESEM, all factors may be linked to all indicators, much like in exploratory factor analysis (EFA). Hence, the ESEM procedure allows us to estimate cross-loadings [37], which provide additional information on the theoretical validity of the scale. The possibility of observing the cross-loadings and the flexibility offered by ESEM give rise to the assumption that ESEM may provide a more appropriate framework for the analysis of the factor structures of psychological inventories designed to measure some complex constructs of individual differences, such as personality traits and motivational factors [38, 39].

Notably, the ESEM procedure covers target rotation, which allows the researcher to have a priori control over the hypothesised factor structure by assuming the cross-loadings to be, if possible, close to zero—but not equal to zero, as in CFA. Therefore, it is practical to rely on ESEM in a confirmatory manner (in contrast to EFA, which is typically used to explore the potential theoretical structures of a construct), i.e., as a way of validating the theoretical structure of a given construct.

The ESEM procedure has also been successfully applied to the SCS. A previous study indicates that the single-bifactor model yields a better fit to the data with ESEM than with CFA [40]. These findings are consistent with an earlier study by Hupfeld and Ruffieux [41], which proved that for analysing the SCS, ESEM yields a better fit to the data than CFA.

Furthermore, some research findings indicate that a bifactor model provides a superior theoretical conceptualisation of self-compassion than that yielded by various other higher-order models. This is because various behaviours related to the content of individual scale items may be considered representations of the general level of self-compassion and its specific components [16, 27]. By these terms, the ESEM procedure appears to be the evidently more

appropriate analytical approach. Most importantly, it permits estimation of both the general and the unique relationship between scale items (i.e., establishing the relationships between specific item groups and both the general self-compassion factor and its specific subcomponents) and encourages examination of the highly sophisticated interaction within the system between items and their cross-loadings. In such circumstances, the general factor(s) in bifactor ESEM models are typically specified similarly as in CFA models, i.e., assuming no cross-loadings between the factors. However, the specific factors are specified as ESEM factors, i.e., the cross-loadings between specific factors are allowed to vary and are not necessarily equal to zero [16].

### 1.3. Self-compassion vs. mental health and well-being

Both self-compassion and mindfulness have turned out to be highly significant variables for mental health, and for depression in particular [42]. However, some studies point to self-compassion playing a more influential role than mindfulness [43]. In addition, patients suffering from depression have lower self-compassion scores than healthy individuals [44]. Research indicates that self-compassion is also significant for anxiety disorders [45–47]. The mitigating impact of self-compassion on social anxiety can be effected via reductions in shame-proneness and irrational beliefs [48].

The breakdown of self-compassion into positive and negative components also seems significant here. Meta-analysis studies show that the positive elements of self-compassion are negatively correlated with psychopathology—which confirms the protective effect of these elements—while the negative elements are positively correlated with psychopathology. In addition, tests comparing the strength of the relationships between the different positive and negative components of self-compassion and psychopathology show that the negative indicators are much more strongly linked to mental health problems than the positive indicators [10]. Furthermore, studies limited to depression report that the three negative components of self-compassion consistently translate to differences between three groups: individuals without depressive symptoms, individuals with depressive syndromes, and individuals with severe depressive disorders [49].

A large meta-analysis of 79 studies confirms significant links between self-compassion and well-being at a level of r = 0.47 [50]. However, there is a stronger relationship between self-compassion and cognitive and mental well-being than between self-compassion and affective well-being. Self-compassion is a significant variable that is predictive of well-being among different groups experiencing various difficulties, including LGBT individuals [51], youth [52], first-year students [53], parents of children with autism [54], athletes [55, 56], women struggling with infertility [57], people living with HIV [58], men who experienced maltreatment as children [59], substance addicts [60], and even chocolate addicts [61].

### 1.4. Research aims and stages of adaptation of Kristin Neff's SCS

This study adapts the SCS by Kristin Neff in keeping with methodological guidelines and instructions obtained from the authors of the test. One of the reasons for the adaptation of the SCS and the preparation of a Polish version is the possibility of using the tool in clinical practice in Poland. Having an SCS tool adapted for Poland will also make the SCS available for psychological assistance and as a support tool in the development of self-compassion as a skill among Polish-speaking peoples, as well as for catering to the psychological balance and well-being of specific individuals seeking support for various life situations. Furthermore, we decided to use the procedures proposed by Neff et al. [16], bearing in mind the importance of replicating their results using slightly different samples. Most of the samples assessed in the

Neff et al. research can be considered as either western, educated, industrialised, prosperous or democratic, with a few exceptions (e.g., Iran). Therefore, it is essential to present additional evidence of the validity of those results by examining the possibility of their extrapolation into less typical samples (e.g., Poland).

The adaptation and preparation of the Polish version of the SCS tool comprised two main parts: (1) cultural validation of the SCS, and (2) psychometric analyses. The procedure for accepting individual statements was performed per the standards set by the authors of the original version of the tool.

Cultural validation of the SCS was performed by having the scale questions translated from English into Polish by qualified translators, followed by a back-translation. In the process of adaptation, the task of determining equivalence in terms of content and functionality was also undertaken to ensure that the Polish version of the scale corresponds to the original version in terms of the graphic form of the tool, the number and order of statements, completion of the instructions, and the procedure for conducting the survey and providing answers to the scale questions [62].

Subsequently, the psychometric properties of the tool were analysed in two successive studies: Study I, and Study II. We decided to conduct two independent studies to increase the validity of our results and formulate highly robust conclusions regarding the stability of the theoretical structure of self-compassion among Polish samples.

## 2. Study I

### 2.1. Methods

**2.1.1. Participants and procedure.** The initial number of participants in this study was 802. However, some data was excluded from the sample because some respondents provided incomplete SCS data. The data was gathered using the snowball sampling method, and a total of 645 participants were included in the final study data. The study participants were predominantly females ($N$ = 401, 62.17%) aged 14–90 ($M$ = 29.26, $SD$ = 12.96). The sample consisted mostly of students at the University of Silesia in Katowice and the Nicolaus Copernicus University in Toruń, and their friends. All participants were Polish residents and no incentive was offered for participation in the study. The participants were informed of the purpose of the study and participated voluntarily. The survey inventory was provided in paper format and electronic format (via an online survey module). The authors of this study requested and obtained official consent to adapt the SCS for Poland. The planned procedure was also approved by the Ethics Committee of the Faculty of Philosophy and Social Sciences of the Nicolaus Copernicus University in Toruń (Approval No. 6/2020).

**2.1.2. Measures.** The SCS is a 26-item tool comprising six subscales of the components of the studied variable. Three of the six subscales are positive: *Self-kindness* (e.g., 'I try to be loving towards myself when I'm feeling emotional pain'); *Common Humanity* (e.g., 'When things are going badly for me, I see the difficulties as part of life that everyone goes through'); and *Mindfulness* (e.g., 'When something upsets me I try to keep my emotions in balance'). The other three subscales are negative and are reverse-coded: *Self-judgement* (e.g., 'I'm disapproving and judgemental about my own flaws and inadequacies'); *Isolation* (e.g., 'When I think about my inadequacies, it tends to make me feel more separate and cut off from the rest of the world'); and *Over-identified* (e.g., 'When I'm feeling down I tend to obsess and fixate on everything that's wrong'). Each item is measured on a 5-point Likert scale ranging from 1 (almost never) to 5 (almost always) [2].

The NEO-FFI [63, 64] is a short form of the NEO Personality Inventory (NEO PI-R). The NEO-FFI tool contains 60 items, and five scales yield scores for the following personality traits:

neuroticism, extraversion, openness to experience, agreeableness and conscientiousness. Each scale comprises 12 items scaled in such a way that higher scores reflect the presence of the traits under evaluation. Participants are asked to respond on a 5-point Likert scale ranging from 0 (strongly disagree) to 4 (strongly agree). In this study, the reliability of the NEO-FFI was satisfactory (Neuroticism: $\alpha = 0.81$, Extraversion: $\alpha = 0.81$, Openness to Experience: $\alpha = 0.72$, Agreeableness: $\alpha = 0.71$, and Conscientiousness: $\alpha = 0.80$).

The Self-Esteem Scale (SES) [65, 66] is a unidimensional tool that enables assessment of a general level of self-esteem construed as a relatively permanent conscious—positive or negative—attitude towards oneself. It comprises ten items rated on a four-point scale on which respondents are required to indicate the degree to which they agree with the statements. High scores indicate high levels of self-esteem. The reliability of the SES tool in this study was satisfactory ($\alpha = 0.79$).

**2.1.3. Analyses.** Study I is focused on the factor structure of the Polish adaptation of the SCS. As proposed by Neff et al. [16], we used both CFA and ESEM.

In this study, the weighted least squares mean-adjusted and variance-adjusted (WLSMV) estimator was used to estimate the parameters of the models because it is suitable for ordered-categorical items with five or fewer answer options [65, 66]. All factor loadings were fully standardised, and for indicators of adequacy of fit, we used a comparative fit index (CFI) and the Tucker–Lewis index (TLI), >0.90, the root mean square error of approximation (RMSEA), <0.08 (with its 90% confidence intervals), and the weighted root mean square residual (WRMR), <1.0. We also report the chi-square values, together with their statistical significance.

We estimated the reliability of the scales using Cronbach's alpha coefficient ($\alpha$), and the McDonald's omega hierarchical coefficient for the general factor ($\omega H$) and subscales ($\omega S$). Furthermore, considering the slight differences in the fit of the six-factor and single-bifactor models, the explained common variance index (ECV) was calculated [67]. The ECV index is considered the best measure of the degree of unidimensionality. It provides information on the proportion of the common variance attributable to the general factor. An ECV value above 0.60 indicates, at least, substantial one-dimensional nature of the tool, while values above 0.90 indicate strict unidimensionality of the test [68].

Consistent with a previous study by Neff [2], we examined gender differences in the SCS scores. We assumed that there would be slight differences in the self-compassion scores of men and women, with women having lower overall self-compassion and mindfulness scores and men having lower levels of self-judgement, isolation, and overidentification. To test for gender differences, we used the Student's t-test for independent samples. The choice of the t-test was dictated by the presence of two comparative groups and the shape of the distribution of the variables in both groups, which did not differ significantly from the normal distribution in terms of skewness (−0.75 to 0.04) and kurtosis (−0.58 to 0.89). The values of the Shapiro–Wilk test of normality indicated differences between the observed distributions and the bell curve (in all cases, p < 0.001). However, close examination of the histograms, skewness and kurtosis showed no deviations from normality in either gender group. Therefore, the results of the Shapiro–Wilk test must have been distorted by the number of participants in the study.

Finally, we examined the discriminant validity of the SCS. We hypothesised that the SCS would be moderately correlated with measures of personality and self-esteem and assumed that the SCS would share some common variance with these measures while remaining a separate theoretical construct. Pearson's correlation coefficient was used to test the hypotheses.

CFA and ESEM were performed using Mplus 7.4 [69], while other analyses were conducted using JASP 0.11.1 [70].

**Table 1. Goodness-of-fit indices for CFA and ESEM models.**

|  | $\chi^2$ (df) | CFI | TLI | RMSEA | 90% CI RMSEA | WRMR |
|---|---|---|---|---|---|---|
| *CFA* |  |  |  |  |  |  |
| Two-factor model | 2497.38*** (298) | .81 | .79 | .11 | .103–.111 | 2.42 |
| Six-factor model | 1576.40*** (284) | .89 | .87 | .08 | .080–.088 | 1.80 |
| Single-bifactor model | 1873.81*** (273) | .86 | .83 | .10 | .091–.099 | 2.26 |
| Two-bifactor model | 1353.30*** (272) | .91 | .89 | .08 | .074–.083 | 1.67 |
| *ESEM* |  |  |  |  |  |  |
| Two-factor model | 2044.15*** (274) | .85 | .82 | .10 | .096–.104 | 1.84 |
| Six-factor model | 429.07*** (184) | .98 | .96 | .05 | .040–.051 | .60 |
| Single-bifactor model | 288.65*** (164) | .99 | .98 | .03 | .028–.041 | .46 |
| Two-bifactor model | 260.78*** (157) | .99 | .98 | .03 | .025–.039 | .43 |

*** $p < .001$

## 2.2. Results

**2.2.1. Factor structure of the Self-Compassion Scale (SCS).**   Consistent with previous studies [16], our results show the inadequacy of the unidimensional model: $\chi^2$ (299) = 3518.07, $p < 0.001$, CFI = 0.72, TLI = 0.69, RMSEA = 0.13 [90% CI: 0.125–0.133], WRMR = 3.31. Therefore, we decided to test four more complex models: the two-factor model, six-factor model, single-bifactor model and the correlated two-bifactor model.

Table 1 presents the model fit indices for both the CFA and ESEM analyses for these models. The two-factor correlated model yielded an inadequate fit, with both the CFA approach and ESEM estimation. Hence, we rejected this solution. Furthermore, the six-factor solution achieved borderline fit with the CFA model, while the ESEM model achieved satisfactory fit. Hence, we accept the latter solution as more accurate. These results appear consistent with the findings of previous studies [16, 71], providing further evidence of a specific pattern of differences between CFA and the ESEM models and indicating that the ESEM model better represents the correlations between factors.

Close analysis of the two-factor and six-factor models revealed moderate covariance between the factors in each solution. Therefore, in keeping with the ideas put forward by Neff et al. [16], we decided to also test two-bifactor models. We examined both models to determine whether one of the two solutions can capture meaningful commonalities between factors, providing a superior representation of the data.

The single-bifactor model (Table 1), comprising one general factor (representing self-compassion) and six specific factors, yielded a rather unsatisfactory fit for CFA. Again, for ESEM, the proposed model had a very satisfactory fit. The test results for the correlated two-bifactor model—which comprised two general factors representing compassionate self-responding (CS) and reduced uncompassionate self-responding (RUS) and six specific factors—also indicate borderline fit for the CFA model and satisfactory fit for the ESEM models.

Table 2 presents the standardised factor loadings for the six-factor model. The CFA factor loadings estimated for the six-factor model were well-defined (average factor loading: $|M_\lambda|$ = 0.70). However, the correlations between the factors were relatively high (average factor correlation: $M_r$ = 0.58), which indicates a significant amount of variance shared between the standardised factors. In the ESEM model, factor loadings were lower (average factor correlation: $|M_\lambda|$ = 0.54) than in the CFA model, and correlations between factors were systematically lower (average factor correlation: $M_r$ = 0.31). Furthermore, some cross-loadings, $\geq$0.32 [72], were found for two self-kindness items on self-judgement and mindfulness, one mindfulness

**Table 2. Standardised factor loadings for the six-factor model: CFA and ESEM solutions.**

| | CFA | ESEM | | | | | |
|---|---|---|---|---|---|---|---|
| | SF (λ) | SK (λ) | CH (λ) | MI (λ) | SJ (λ) | IS (λ) | OI (λ) |
| *Self-kindness* | | | | | | | |
| item 5 | .67 | **.71** | .03 | .04 | *.01* | -.10 | .11 |
| item 12 | .55 | **.70** | .08 | -.14 | *-.04* | .11 | -.09 |
| item 19 | .81 | **.77** | .07 | .10 | *.00* | *.03* | *.03* |
| item 23 | .75 | **.33** | .10 | .21 | .39 | .13 | -.18 |
| item 26 | .73 | **.42** | .08 | .34 | .36 | -.09 | -.14 |
| *Common humanity* | | | | | | | |
| item 3 | .62 | .19 | **.36** | *.04* | -.15 | *.06* | .22 |
| item 7 | .70 | -.11 | **.88** | *-.02* | *.04* | *-.01* | -.10 |
| item 10 | .73 | -.07 | **.94** | *-.05* | *-.02* | *.03* | -.09 |
| item 15 | .84 | .26 | **.32** | .26 | *-.04* | *.08* | .11 |
| *Mindfulness* | | | | | | | |
| item 9 | .72 | .11 | .14 | **.51** | *-.06* | *.02* | .28 |
| item 14 | .65 | .05 | .06 | **.65** | *-.08* | *.01* | .22 |
| item 17 | .61 | .24 | .20 | **.18** | *.03* | -.06 | .23 |
| item 22 | .59 | .41 | .18 | **.26** | *-.08* | *.03* | *-.04* |
| *Self-judgment* | | | | | | | |
| item 1 | .71 | .00 | .12 | -.05 | **.73** | -.16 | .21 |
| item 8 | .71 | .07 | *.02* | -.17 | **.65** | *.00* | .21 |
| item 11 | .65 | .13 | -.07 | .07 | **.48** | .15 | -.01 |
| item 16 | .74 | .07 | -.05 | .06 | **.67** | .13 | -.05 |
| item 21 | .63 | .25 | -.12 | -.19 | **.33** | .30 | .10 |
| *Isolation* | | | | | | | |
| item 4 | .75 | .05 | .04 | -.08 | .19 | **.38** | .32 |
| item 13 | .73 | -.05 | .08 | *-.01* | *-.01* | **.82** | *.02* |
| item 18 | .66 | -.09 | .02 | .04 | *.01* | **.71** | *.05* |
| item 25 | .74 | .17 | *.01* | -.05 | .08 | **.54** | .13 |
| *Over-identification* | | | | | | | |
| item 2 | .76 | -.02 | .06 | .05 | .26 | *.04* | **.64** |
| item 6 | .73 | .10 | .06 | .04 | .22 | .35 | **.32** |
| item 20 | .66 | .23 | *-.01* | .39 | .09 | .20 | **.32** |
| item 24 | .74 | .08 | -.04 | .37 | .11 | .25 | **.34** |

*Notes*. CFA—Confirmatory factor analysis; ESEM—Exploratory structural equation modeling; SF—specific factor loading (in CFA, when cross-loadings were constrained to zero); λ—standardized factor loadings.

For SCS factors: SK—self-kindness; CH—common humanity; MI—mindfulness; SJ—self-judgment; IS—isolation; OI—over-identification.

In the ESEM model, target factor loadings are bolded. Non-significant factor loadings are in italics.

item on self-kindness, one isolation item on overidentification, and two overidentification items on mindfulness and isolation.

The parameter estimates for the single-bifactor model are presented in Table 3. The results for the CFA reveal a well-defined general factor (average factor loading: $|M_\lambda|$ = 0.53). Furthermore, six specific factors exhibit a moderate to high degree of specificity, with the highest being for the *common humanity* factor ($|M_\lambda|$ = 0.56), and the lowest being for *overidentification* ($|M_\lambda|$ = 0.34) and *mindfulness* ($|M_\lambda|$ = 0.35). The remaining factors exhibit a moderate degree of specificity (*self-kindness*: |Mλ| = 0.45, *self-judgement*: |Mλ| = 0.44, *isolation*: |Mλ| = 0.46).

**Table 3. Standardised factor loadings for single-bifactor model: CFA and ESEM solutions.**

| | CFA | | ESEM | | | | | | |
|---|---|---|---|---|---|---|---|---|---|
| | GF (λ) | SF (λ) | GF (λ) | SK (λ) | CH (λ) | MI (λ) | SJ (λ) | IS (λ) | OI (λ) |
| **Self-kindness** | | | | | | | | | |
| item 5 | .48 | .60 | **.49** | **.56** | .11 | *-.02* | *-.04* | -.14 | *.02* |
| item 12 | .37 | .54 | **.37** | **.56** | .13 | -.12 | *-.01* | .03 | -.16 |
| item 19 | .61 | .60 | **.58** | **.61** | .14 | *.05* | *-.02* | -.05 | *-.01* |
| item 23 | .66 | .20 | **.59** | **.24** | .08 | .19 | .28 | .06 | -.13 |
| item 26 | .61 | .30 | **.60** | **.25** | *.07* | .28 | .18 | -.15 | -.27 |
| **Common humanity** | | | | | | | | | |
| item 3 | .42 | .35 | **.50** | *.05* | **.27** | -.11 | -.30 | -.09 | -.15 |
| item 7 | .31 | .75 | **.30** | .12 | **.69** | .08 | *-.03* | -.08 | *-.02* |
| item 10 | .33 | .81 | **.29** | .16 | **.86** | *.07* | *-.03* | -.03 | *.03* |
| item 15 | .60 | .32 | **.63** | .09 | **.25** | .13 | -.20 | *-.07* | -.19 |
| **Mindfulness** | | | | | | | | | |
| item 9 | .58 | .56 | **.62** | -.04 | .11 | **.31** | -.18 | -.09 | .18 |
| item 14 | .51 | .53 | **.57** | -.08 | *.04* | **.43** | -.21 | -.10 | .15 |
| item 17 | .48 | .17 | **.51** | .16 | .17 | **.07** | -.08 | -.13 | *.08* |
| item 22 | .56 | .14 | **.39** | .38 | .22 | **.24** | *-.08* | *-.04* | *.07* |
| **Self-judgment** | | | | | | | | | |
| item 1 | .52 | .43 | **.57** | -.04 | *.01* | -.11 | **.48** | -.12 | *.03* |
| item 8 | .52 | .55 | **.52** | .05 | -.05 | -.19 | **.50** | .02 | .12 |
| item 11 | .51 | .35 | **.50** | .02 | -.13 | *.03* | **.35** | .11 | -.08 |
| item 16 | .56 | .50 | **.54** | .00 | -.12 | *.04* | **.51** | .12 | -.05 |
| item 21 | .48 | .35 | **.44** | .19 | -.14 | -.18 | **.33** | .25 | .11 |
| **Isolation** | | | | | | | | | |
| item 4 | .62 | .30 | **.63** | -.08 | -.09 | -.19 | .10 | **.24** | *.07* |
| item 13 | .54 | .62 | **.55** | -.12 | -.05 | *-.03* | *.05* | **.60** | *.00* |
| item 18 | .48 | .57 | **.47** | -.12 | -.09 | *.01* | .07 | **.52** | .11 |
| item 25 | .61 | .33 | **.56** | .10 | -.06 | -.09 | .12 | **.40** | .13 |
| **Over-identification** | | | | | | | | | |
| item 2 | .70 | .25 | **.74** | -.17 | -.07 | -.16 | *.05* | -.03 | **.26** |
| item 6 | .79 | *.06* | **.75** | -.04 | -.05 | -.09 | .11 | .21 | **.09** |
| item 20 | .44 | .56 | **.45** | -.21 | -.05 | .26 | *.05* | .13 | **.45** |
| item 24 | .61 | .49 | **.60** | -.12 | -.09 | .21 | *.04* | .15 | **.36** |

*Notes*. CFA—Confirmatory factor analysis; ESEM—Exploratory structural equation modeling; SF—specific factor loading (in CFA, when cross-loadings were constrained to zero); GF—general factor; λ—standardized factor loadings.

For specific factors: SK—self-kindness; CH—common humanity; MI—mindfulness; SJ—self-judgment; IS—isolation; OI—over-identification.

Target factor loadings are bolded. Non-significant factor loadings are in italics.

The results of ESEM also provided a well-defined general factor, with a similar level of factor loadings (average factor loading: $|M_\lambda| = 0.53$). Six specific factors had a diverse degree of specificity: high for *common humanity* ($|M_\lambda| = 0.52$); moderate for *self-kindness* ($|M_\lambda| = 0.44$), *self-judgement* ($M_\lambda = 0.43$) and *isolation* ($|M_\lambda| = 0.44$); and low for *overidentification* ($|M_\lambda| = 0.29$) and *mindfulness* ($|M_\lambda| = 0.26$). Compared to the six-factor model, the levels of cross-loadings were lower, providing superior representation of the factor structure of the scale.

The parameter estimates for the two-bifactor model are presented in Table 4. The results of CFA reveal two weakly defined general factors, with rather moderate factor loadings for CS

**Table 4. Standardised factor loadings for the two-bifactor model: CFA and ESEM solutions.**

| | CFA | | | ESEM | | | | | | | |
|---|---|---|---|---|---|---|---|---|---|---|---|
| | CS (λ) | RUS (λ) | SF (λ) | CS (λ) | RUS (λ) | SK (λ) | CH (λ) | MI (λ) | SJ (λ) | IS (λ) | OI (λ) |
| *Self-kindness* | | | | | | | | | | | |
| item 5 | .58 | | .49 | **.19** | | **.68** | .22 | *.08* | .11 | *-.02* | .15 |
| item 12 | .47 | | .45 | **.17** | | **.63** | .27 | *-.07* | .10 | .13 | *-.07* |
| item 19 | .72 | | .49 | **.26** | | **.73** | .25 | .15 | .16 | .09 | .13 |
| item 23 | .76 | | -.09 | **.23** | | **.33** | .37 | .21 | .50 | .18 | *.04* |
| item 26 | .71 | | .06 | **.33** | | **.34** | .35 | *.17* | .48 | -.09 | .07 |
| *Common humanity* | | | | | | | | | | | |
| item 3 | .48 | | .27 | **.13** | | .23 | **.55** | *-.09* | *-.03* | .10 | .25 |
| item 7 | .38 | | .71 | **-.24** | | .28 | **.63** | .28 | *.01* | *.01* | *-.02* |
| item 10 | .40 | | .78 | **-.34** | | .34 | **.59** | .34 | *-.04* | .07 | *-.05* |
| item 15 | .67 | | .23 | **.28** | | .24 | **.21** | *.10* | .14 | .10 | .22 |
| *Mindfulness* | | | | | | | | | | | |
| item 9 | .64 | | .66 | **.27** | | .13 | .35 | **.38** | .09 | .09 | .43 |
| item 14 | .57 | | .35 | **.37** | | *.00* | .32 | **.44** | .08 | .05 | .41 |
| item 17 | .58 | | .04 | **.10** | | .32 | .32 | **.16** | .11 | *.01* | .31 |
| item 22 | .56 | | -.03 | **.19** | | .44 | .27 | **.34** | *.03* | *.04* | *.08* |
| *Self-judgment* | | | | | | | | | | | |
| item 1 | | .60 | .37 | | **-.13** | .18 | .12 | *.04* | **.62** | .12 | .30 |
| item 8 | | .58 | .51 | | **-.28** | .24 | -.04 | *.03* | **.59** | .35 | .28 |
| item 11 | | .56 | .27 | | **.25** | .16 | *.04* | *.01* | **.59** | .06 | .18 |
| item 16 | | .61 | .43 | | **.09** | .15 | *.02* | .08 | **.70** | .18 | .17 |
| item 21 | | .54 | .27 | | **.07** | .32 | -.12 | *-.02* | **.41** | .38 | .17 |
| *Isolation* | | | | | | | | | | | |
| item 4 | | .67 | .20 | | **.21** | .14 | .13 | *-.09* | .36 | **.37** | .41 |
| item 13 | | .59. | .58 | | **.43** | *.01* | .13 | *.02* | .31 | **.58** | .20 |
| item 18 | | .53 | .52 | | **.29** | *-.02* | *.04* | .12 | .26 | **.59** | .21 |
| item 25 | | .65 | .25 | | **.28** | .26 | *.05* | *.05* | .30 | **.46** | .26 |
| *Over-identification* | | | | | | | | | | | |
| item 2 | | .75 | .15 | | **.06** | .14 | .17 | *.02* | .33 | .23 | **.68** |
| item 6 | | .85 | -.13 | | **.25** | .21 | .19 | *.02* | .41 | .34 | **.47** |
| item 20 | | .50 | .48 | | **.20** | *-.07* | *-.01* | .42 | .18 | .18 | **.49** |
| item 24 | | .66 | .45 | | **.27** | *.06* | *.04* | .36 | .26 | .21 | **.53** |

*Notes*. CFA—Confirmatory factor analysis; ESEM—Exploratory structural equation modeling; SF—specific factor loading (in CFA, when cross-loadings were constrained to zero); λ—standardized factor loadings.

For general factors: CS—Compassionate self-responding; RUS—Reduced uncompassionate self-responding.

For specific factors: SK—self-kindness; CH—common humanity; MI—mindfulness; SJ—self-judgment; IS—isolation; OI—over-identification.

Target factor loadings are bolded. Non-significant factor loadings are in italics.

($|M_\lambda| = 0.58$) and RUS ($|M_\lambda| = 0.63$). Furthermore, six specific factors exhibited a low to moderate degree of specificity: high for *common humanity* ($|M_\lambda| = 0.78$); moderate for *self-kindness* ($|M_\lambda| = 0.32$), *self-judgement* ($|M_\lambda| = 0.37$) and *isolation* ($|M_\lambda| = 0.39$); and low for *overidentification* ($|M_\lambda| = 0.30$) and *mindfulness* ($|M_\lambda| = 0.27$). However, some of the factor loadings were unexpectedly negative, which complicates the interpretation of specific factors. Both general factors were highly correlated ($r = 0.65$, $p < 0.001$).

**Table 5. Reliability estimates for the SCS.**

|  | ω$_H$ | ω$_S$ | α | ECV |
|---|---|---|---|---|
| General | .83 | - | .66 | .89 |
| Self-kindness | - | .03 | .80 | - |
| Common humanity | - | .02 | .74 | - |
| Mindfulness | - | .01 | .68 | - |
| Self-judgment | - | .01 | .78 | - |
| Isolation | - | .01 | .77 | - |
| Over-identification | - | .01 | .77 | - |

*Notes.* ω$_H$—McDonald's coefficient omega hierarchical; ω$_S$—McDonald's coefficient omega hierarchical subscale; ω$_t$ _McDonald's omega total; α—Cronbach's internal consistency (alpha coefficient)

The results of ESEM confirmed the rather uncertain status of the two general factors, with notably low factor loadings for CS ($|M_\lambda| = 0.24$) and RUS ($|M_\lambda| = 0.22$). Some of the factor loadings were negative, and six specific factors exhibited a rather moderate degree of specificity: *common humanity* ($|M_\lambda| = 0.50$), *self-kindness* ($|M_\lambda| = 0.54$), *self-judgement* ($|M_\lambda| = 0.58$), *isolation* ($|M_\lambda| = 0.50$), *overidentification* ($|M_\lambda| = 0.54$), *mindfulness* ($|M_\lambda| = 0.33$). Compared to the six-factor solution, the cross-loadings were higher. Again, some of the factor loadings were unexpectedly negative, which complicated the interpretation of the specific factors. Both general factors were moderately correlated ($r = 0.36$, $p = 0.005$).

In summary, the six-factor correlated model and the single-bifactor model provide a more meaningful representation of the theoretical structure of the scale. Although the two-bifactor model appears to yield a satisfactory fit, a close examination of the parameter estimates inclined us to reject it as difficult to interpret (both when general and specific factors are taken into account).

**2.2.2. Reliability of the Self-Compassion Scale (SCS).** We estimated the reliability indices for the SCS tool. The values of the reliability indices and the ECV index are presented in Table 5.

The analysis indicates acceptable and satisfactory reliability of the general scores, which are lowest and borderline for Cronbach's alpha and highest for the McDonald's omega hierarchical coefficient. Furthermore, the ECV index confirmed that the unidimensionality of the scale is substantial, indicating that a considerable amount of the variance is attributable to the main factor.

Most of the subscales yielded a satisfactory level of reliability, with the mindfulness subscale having a borderline—and yet acceptable—level of reliability. However, the McDonald's omega hierarchical coefficient indicates that, after partitioning out the variance of the main factor, very little common variance is attributable to the specific factors, which reduces the reliability of the specific factor scores. Nevertheless, the results appear similar to those obtained by Neff et al. [16], indicating that, for the bifactor model, 95% of the reliable variance in the scores is attributable to the general factor and only 5% is attributable to the specific factors.

**2.2.3. Differences between men and women in self-compassion.** The broad scores of men and women, together with the gender differences between scores, are presented in Table 6.

Results of the *t*-test for the individual groups reveal statistically significant differences in general self-compassion, with higher scores for men. Furthermore, women had significantly higher scores in self-judgement, isolation, and overidentification, while men scored higher in

**Table 6. Mean scores and gender differences in self-compassion.**

|  | Total sample | | Men | | Women | | Gender differences | | |
|---|---|---|---|---|---|---|---|---|---|
|  | *M* | *SD* | *M* | *SD* | *M* | *SD* | *t* (627) | *p* | Cohen's *d* |
| General | 2.93 | .65 | 3.08 | .57 | 2.84 | .67 | 4.58 | < .001 | .38 |
| Self-kindness | 2.96 | .87 | 2.98 | .85 | 2.95 | .88 | .47 | .641 | .04 |
| Common humanity | 3.07 | .86 | 3.01 | .84 | 3.10 | .86 | -1.25 | .212 | -.10 |
| Mindfulness | 3.10 | .80 | 3.29 | .72 | 2.99 | .82 | 4.61 | < .001 | .38 |
| Self-judgment | 3.18 | .83 | 3.06 | .81 | 3.24 | .84 | -2.71 | .007 | -.23 |
| Isolation | 3.14 | .96 | 2.92 | .95 | 3.27 | .94 | -4.52 | < .001 | -.38 |
| Over-identification | 3.26 | .98 | 2.83 | .95 | 3.51 | .93 | -8.73 | < .001 | -.72 |

*Note*. General self-compassion scores were calculated by reverse coding the self-judgment, isolation, and over-identification items then summing all six subscale means. For sake of simplicity in interpretation of subscales, scores in self-judgment, isolation, and over-identification were calculated based on non-reverse coded answers.

mindfulness. The results were consistent with previously observed gender differences in self-compassion [2].

**2.2.4. Discriminant validity of the SCS.** Correlations between the SCS and measures of personality and self-esteem are presented in Table 7.

It was found that neuroticism is the one personality trait with a consistent, moderate and strong correlation with self-compassion and its facets. A significantly high level of neuroticism is linked to significantly high levels of self-judgement, isolation, and overidentification. Furthermore, people with high levels of neuroticism tend to have low scores on general self-compassion, self-kindness, common humanity, and mindfulness. Furthermore, high extraversion is linked to high general self-compassion and a strong tendency towards self-kindness, common humanity, and mindfulness. Extraversion is negatively correlated with self-judgement, isolation and overidentification. A high level of openness is related to high scores in common humanity and self-judgement. People with high levels of agreeableness exhibit high levels of self-compassion, self-kindness, and common humanity and have only a slight tendency towards isolation. Finally, conscientiousness is positively related to general self-compassion, self-kindness, common humanity, and mindfulness, but it is negatively related to self-judgement, isolation, and overidentification. However, the relationships between self-compassion, extraversion, openness, agreeableness, and conscientiousness traits are, however, very weak.

**Table 7. Correlations between the SCS, the Big Five personality dimensions (NEO-FFI), and Self-Esteem (SES), measured using Pearson's correlation coefficient.**

|  | SC | SK | CH | MI | SJ | IS | OI |
|---|---|---|---|---|---|---|---|
| Neuroticism | -.74*** | -.42*** | -.35*** | -.53*** | .52*** | .69*** | .74*** |
| Extraversion | .34*** | .34*** | .23*** | .20*** | -.21*** | -.28*** | -.21*** |
| Openness | .01 | .07 | .10* | .06 | .08* | .03 | .04 |
| Agreeableness | .14*** | .24*** | .12** | .05 | -.07 | -.11** | -.01 |
| Conscientiousness | .23*** | .16*** | .15*** | .14*** | -.08* | -.28*** | -.18*** |
| Rosenberg's self-esteem | .58*** | .52*** | .34*** | .37*** | -.40*** | -.48*** | -.41*** |

*Note*. General self-compassion scores were calculated by reverse coding the self-judgment, isolation, and over-identification items then summing all six subscale means. For sake of simplicity in interpretation of subscales, scores in self-judgment, isolation, and over-identification were calculated based on non-reverse coded answers.

SC—general self-compassion; SK—self-kindness; CH—common humanity; MI—mindfulness; SJ—self-judgment; IS—isolation; OI—over-identification.

* $p < .05$,

** $p < .01$,

*** $p < .001$

Self-esteem is positively and strongly correlated to general self-compassion, and a high level of self-esteem is related to a strong tendency towards self-kindness, common humanity, and mindfulness. Furthermore, self-esteem is negatively correlated with self-judgement, isolation, and overidentification.

## 3. Study II

### 3.1. Methods

**3.1.1. Participants and procedure.** The initial number of participants in this study was 696. However, some data was excluded because of missing SCS data, resulting in only 688 participants being included in the final sample. The study participants were predominantly females ($N$ = 460, 67.35%) aged 15–69 ($M$ = 25.62, $SD$ = 9.78). All participants were Polish residents and no incentive was offered for participation in the study.

**3.1.2. Measures.** The SCS is described in Study I.

The State–Trait Anxiety Inventory (STAI) [59, 60] is a tool designed for studying state anxiety (understood as a transient and situationally determined state in an individual) and trait anxiety (understood as a relatively constant personality trait). The STAI has two subscales, one of which is used to measure state anxiety and the other to measure trait anxiety. Only the subscale for measuring trait anxiety was used in this study. Each subscale comprises 20 items to which the study subject responds by selecting one of four answers on a scale of 1 (almost never) to 4 (almost always). The higher the score, the greater the intensity of the trait. The reliability of the STAI trait anxiety scale was satisfactory ($\alpha$ = 0.89).

The Trait Emotional Intelligence Questionnaire (TEIQue) [73, 74] is a tool comprised of 33 statements describing different behaviours. On a five-point scale (graded from 'strongly disagree' to 'strongly agree'), the respondent indicates the extent to which they identify with each statement. The broad score is obtained by adding the points for the answers to each question. In this study, the reliability of the TEIQue tool was satisfactory ($\alpha$ = 0.89).

The Beck Depression Inventory [75, 76] is a tool for evaluating the symptoms of depression. The questionnaire comprises 21 items. After reading each scale item, the respondent selects one of four statements that most accurately describe how they have been feeling over the preceding two weeks. The responses are rated from 0 to 3, reflecting increasing severity of the condition in that specific area. The broad score is obtained by adding the points for the answers to each question. In this study, the reliability of the tool was satisfactory ($\alpha$ = 0.95).

The Satisfaction with Life Scale [77, 78] is a tool for measuring global life satisfaction. The scale consists of five statements, which respondents rate on a 7-point scale, from 1 (completely disagree) to 7 (completely agree). The broad score is obtained by adding the score given to each statement. In this study, the reliability of this tool was satisfactory ($\alpha$ = 0.85).

**3.1.3. Analyses.** We tested the discriminant validity of the SCS using Pearson's correlation coefficient, as in Study I. We hypothesised that the SCS would be moderately correlated with measures of *satisfaction with life*, trait anxiety, depression, and emotional intelligence. As in Study I, we assumed that the SCS would share some common variance with these measures while remaining a separate theoretical construct.

CFA and ESEM were performed using Mplus 7.4 [69], and the other analyses were conducted using JASP 0.11.1 [70].

### 3.2. Results

**3.2.1. Factor structure of the SCS.** First, we tested the theoretical structure of the SCS using CFA and ESEM. We considered the two models that yielded the best fit in Study I: the

**Table 8. Correlations between the SCS, *satisfaction with life* (SWLS), emotional intelligence (INTE), depressive symptoms (BDI), and trait anxiety (STAI).**

|                        | SC       | SK       | CH       | MI       | SJ       | IS       | OI       |
|------------------------|----------|----------|----------|----------|----------|----------|----------|
| Satisfaction with life | .48***   | .48***   | .37***   | .32***   | -.35***  | -.37***  | -.30***  |
| Emotional intelligence | .38***   | .42***   | .38***   | .35***   | -.17***  | -.24***  | -.18***  |
| Depressive symptoms    | -.49***  | -.38***  | -.31***  | -.25***  | .42***   | .42*     | .39***   |
| Trait anxiety          | -.74***  | -.55***  | -.38***  | -.51***  | .57***   | .64***   | .67***   |

SC—general self-compssion; SK—self-kindness; CH—common humanity; MI—mindfulness; SJ—self-judgment; IS—isolation; OI—over-identification.

* $p < .05$,

** $p < .01$,

*** $p < .001$

six-factor correlated model and the single-bifactor model. As in Study I, the WLSMV estimation method was used, along with the same cut-off values for fit indices.

A close examination of the fit indices reveals that the six-factor model has an unsatisfactory fit in CFA: $\chi^2$ (284) = 1970.55, $p < 0.001$, CFI = 0.89, TLI = 0.88, RMSEA = 0.09 [90% CI: 0.089–0.097]. However, the six-factor model achieved a satisfactory fit using ESEM: $\chi^2$ (184) = 561.55, $p < 0.001$, CFI = 0.98, TLI = 0.96, RMSEA = 0.06 [90% CI: 0.049–0.060]. Using the ESEM procedure, the single-bifactor model yielded a satisfactory fit: $\chi^2$ (164) = 442.07, $p < 0.001$, CFI = 0.99, TLI = 0.97, RMSEA = 0.04 [90% CI: 0.044–0.055], which is superior to that achieved using CFA: $\chi^2$ (273) = 2626.21, $p < 0.001$, CFI = 0.85, TLI = 0.82, RMSEA = 0.11 [90% CI: 0.108–0.116]. Furthermore, comparing the fit of the bifactor and six-factor models achieved using the ESEM procedure, the bifactor model achieved a better fit. Therefore, we conclude that the single-bifactor solution provides a slightly better representation of the data than the six-factor model does.

**3.2.2. Discriminant validity of the SCS.** Pearson's correlation coefficient was used to test the discriminant validity hypotheses. The correlations between the measures are presented in Table 8.

It was found that *satisfaction with life* is positively and moderately correlated with self-compassion and its facets. A high level of *satisfaction with life* is linked to a high level of general self-compassion, self-kindness, common humanity, and mindfulness. Furthermore, a high level of *satisfaction with life* is linked to low levels of self-judgement, isolation, and overidentification.

In addition, we found similar patterns of relationships for correlations between the SCS and emotional intelligence. High emotional intelligence is linked to a high level of self-compassion and a strong tendency towards self-kindness, common humanity, and mindfulness. Furthermore, emotional intelligence is negatively correlated with self-judgement, isolation, and overidentification.

However, self-compassion is also negatively related to depressive symptoms and trait anxiety. The correlations between these variables are mostly moderate or strong. Hence, a high level of depressive symptoms is related to low general self-compassion scores, self-kindness, common humanity, and mindfulness. Furthermore, individuals with high levels of depressive symptoms demonstrate high levels of self-judgement, isolation, and overidentification.

The preceding pattern was also observed for trait anxiety. Trait anxiety is positively related to self-judgement, isolation, and overidentification and negatively related to general self-compassion, self-kindness, common humanity, and mindfulness.

## 4. Discussion

This two-part study confirms the six-factor structure of the SCS, which comprises the three positive self-compassion components (self-kindness, common humanity and mindfulness) and the three negative components (overidentification, isolation and self-judgement). However, the results of analyses using structural equation modelling (SEM) and ESEM indicate that, apart from the six intercorrelated factors, it is possible to identify a single main factor, conceptualised as self-compassion. However, the analyses also indicate that the competitive theoretical models (single-factor, two-factor, and two-bifactor models) have insufficient fit. These results are consistent with the original findings of Kristin Neff [2]. Many replication studies have also confirmed the six-factor structure [13–15]. Our two-part study also replicates the findings of Neff et al. [16] The SCS was also characterised by good theoretical validity with respect to gender differences in terms of self-compassion and its components. From the analyses, while men generally have higher scores in self-compassion and its subcomponent, mindfulness, women exhibit a higher level of negative self-compassion components (i.e., self-judgement, isolation, and overidentification). In a meta-analysis of 71 studies, Yarnell et al. [79] also report generally slightly higher self-compassion scores among men. Interestingly, age was negatively correlated with the magnitude of gender differences, with these differences being smaller among older subjects.

Regarding discriminant validity, the results of our analysis indicate a conceptual distinctiveness of self-compassion with regard to the Big Five personality traits. The greatest similarities in terms of shared common variance, which were found in relation to neuroticism, correlated with self-compassion to a high or moderate degree. As far as the other personality traits are concerned, the SCS correlated moderately or weakly with extraversion and weakly with the other personality traits. Self-compassion also turned out to be a separate construct from self-esteem. The results of the analyses point to moderate relationships between the SCS and its dimensions on the one hand and the broad level of self-esteem on the other.

## 5. Limitations

There are some limitations to our study, one of which is related to the character of the sample. Most of the samples assessed in previous studies [29–32] can be considered as either western, educated, industrialised, prosperous or democratic. Bearing in mind that our study was conducted in Poland, we believe that our research might not be comparable to those conducted previously. Hence, it seems essential to confirm the validity of these results by evaluating the possibility of their extrapolation to other less typical samples. However, our samples were quite large, which makes our results particularly reliable.

Although our study confirms the single-bifactor conceptualisation of self-compassion, we believe that the division between compassion and self-coldness may be applicable in some groups. Indeed, many studies confirm the legitimacy of two-factor analysis, pointing to the significant role of self-coldness [7–10]. Other studies also point to the psychometric legitimacy of analysing a two-factor model while taking into account self-compassion and self-coldness [8,10, 80, 81]. The research cited in this study and the contradictions they contain show that the debate on the factor structure of SCS is highly complex and multi-faceted. However, further research efforts are all the more justified to dispel the doubts that have accumulated and establish whether and in which cases the two-factor structure of the scale is sustainable, as well as in which cases a one-factor solution is preferable. It also seems justified that in subsequent research, the factor structure of a specific group should be tested before analysis.

Furthermore, our findings highlight some limitations of CFA, which is often used as a routine procedure while testing the theoretical structures of psychological constructs. The

research conclusions cited in this study may have resulted from theoretical inadequacies and simplifications generated while using CFA, e.g., fixing cross-loadings as zero. In this regard, our findings demonstrate the superiority of the ESEM bifactor approach over CFA. However, psychometric studies that employ ESEM are still much more unique examples than typical procedures. Hence, we recommend expanding the number of studies that use ESEM procedures in general and conducting more research on self-compassion in clinical samples. Such endeavours should be worthwhile, as they might produce an unambiguous solution to the debate.

## 6. Conclusion

The Polish versions of the 26 (long) forms of the SCS have been validated as reliable instruments for measuring self-compassion in the general population. Given the good psychometric properties of the tool, its use is recommended for the evaluation of self-compassion and its components in both clinical and research environments. Analyses should be based on the results obtained on the six subscales or on the broad score. Due to the lively debate and the emerging analyses supporting factorial analysis of six subscales versus the broad score or two subscales, we recommend that the factor structure of the studied group be analysed each time.

## Supporting information

**S1 Appendix.**
(DOCX)

## Author Contributions

**Conceptualization:** Dagna Kocur, Małgorzata Fopka-Kowalczyk.

**Data curation:** Dagna Kocur, Małgorzata Fopka-Kowalczyk.

**Formal analysis:** Maria Flakus.

**Methodology:** Maria Flakus.

**Resources:** Dagna Kocur.

**Writing – original draft:** Dagna Kocur, Maria Flakus, Małgorzata Fopka-Kowalczyk.

**Writing – review & editing:** Dagna Kocur, Maria Flakus, Małgorzata Fopka-Kowalczyk.

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
