## [Decision Letter · Decision Letter 0]

1 Dec 2021

PONE-D-21-04597

Validity and reliability of the Polish version of the Self-Compassion Scale and its correlates

PLOS ONE

Dear Dr. Kocur,

Thank you for submitting your manuscript to PLOS ONE. After careful consideration, we feel that it has merit but does not fully meet PLOS ONE’s publication criteria as it currently stands. Therefore, we invite you to submit a revised version of the manuscript that addresses the points raised during the review process.

We look forward to receiving your revised manuscript.

Kind regards,

Frantisek Sudzina

Academic Editor

PLOS ONE

Journal Requirements:

2. Please note that PLOS ONE does not copy edit accepted manuscripts (https://journals.plos.org/plosone/s/criteria-for-publication#loc-5). To that effect, please ensure that your submission is free of typos and grammatical errors.

Reviewers' comments:

Reviewer's Responses to Questions

**Comments to the Author**

1. Is the manuscript technically sound, and do the data support the conclusions?

Reviewer #1: Partly

Reviewer #2: Yes

2. Has the statistical analysis been performed appropriately and rigorously? 

Reviewer #1: No

Reviewer #2: Yes

3. Have the authors made all data underlying the findings in their manuscript fully available?

Reviewer #1: Yes

Reviewer #2: Yes

4. Is the manuscript presented in an intelligible fashion and written in standard English?

Reviewer #1: No

Reviewer #2: No

5. Review Comments to the Author

Reviewer #1: Journal: PLOS ONE

Manuscript ID: PONE-D-21-04597

Full title: Validity and reliability of the Polish version of the Self-Compassion Scale and its correlates

The present paper, in two studies, reported the validation of the Polish version of the Self-Compassion Scale which encompassed: (1) testing its factor structure using confirmatory factor analytic (CFA) and exploratory structural equation modeling (ESEM) models; (2) calculating alternative reliability indices; (3) establishing its construct validity in relation to personality traits, self-judgment, depressive symptoms, trait anxiety, life satisfaction and emotional intelligence.

Overall, I believe that the present work has its merits as it tackles an important question. At the same time, I had some questions regarding the theoretical foundations and the conducted analyses while reading the manuscript that might be helpful.

Since there are two alternative approaches to how self-compassion should be understood (Neff with the notion that self-compassion is best represented by one global factor, and Gilbert with the notion that self-compassion is best represented by two global factors), the authors should emphasize that their paper (and the SCS) follows Neff’s operationalization of self-compassion.

The Introduction contains some repetition: the authors first talk about self-compassion and its correlates on page 3 but then bring up roughly the same idea on page 8-9. These two parts could be merged to shorten the Introduction.

While the authors did a good job in presenting the competing views on self-compassion, I feel this could be done in a more balanced way. Currently, the two-factor representation gets more discussion. I would suggest the authors to incorporate the recent work of Neff (2019, 2020).

The description of the SCS appears to be missing.

It is unclear why the authors tested gender differences. What is the importance of these tests? Is there a theoretical reason for conducting them?

While it seems that the authors followed the analytic plan outlined by Neff et al. (2019), they also do not seem to follow it closely. For example, there is no word on whether bifactor models (including on or two global factors) which is an important part of this study and the whole discussion on the most optimal representation of self-compassion.

The various omega indices appear to be mixed up. Although it is true that all are derivatives of McDonald’s original work, the equation of the original omega index (let’s call it composite reliability [CR] for the sake of simplicity) is different from the equation of the omega and omega hierarchical indices that have specifically been developed for bifactor models. This is because they refer to different aspects of reliability: CR refers to the actual reliability of a scale (akin to Cronbach’s alpha), while the other two refer to the amount of reliable variance attributed to the various factors. I suggest the authors to consult the work of McDonald (1970) as well as Rodriguez et al. (2016).

When describing the alternative factor solutions, the authors spend way too much time on the improper solutions. Given the length of the paper, I would suggest to simply report all fit indices in a table, then quickly move to the comparison of the most adequate solutions. Again, consult with Neff et al. (2019) on the structure of how to report these models in a concise manner, and which aspects of the models (factor loadings, cross-loadings, factor correlations) to report. That is, the authors should first focus on the comparison of the first-order solutions, retain one. With the bifactor models, first test whether one or two global factors should be incorporated (focus on the definition of the global factors and the correlations between the two global factors), retain one of the bifactor models and compare this bifactor model to its first-order counterpart.

According to APA guidelines, it is redundant to present detailed results both in tables/figures and in the text.

Factor loading tables: add significance values to all factor loadings instead of using italics, there is plenty of space left.

The degrees of freedoms of the single-bifactor model (unclear if CFA- or ESEM-based model) reported in Study 2 do not match the degrees of freedoms in Study 1 which means that different models were estimated. This gives me a lot of concern about the results.

Where is the limitations section?

There are a few sections in the paper which are not presented in their correct location. For example, the authors should not talk about prior ESEM-based studies and results in the Analyses section. Similarly, the authors should not talk about the ethical aspects of their study in the Present Study section.

Finally, there are a fair few typos and unusual sentence structures throughout the manuscript which made it difficult to understand parts of the manuscript. Thus, I would suggest the authors to ask a colleague (whose native language is English) to read the paper before submission.

References used in this review:

McDonald, R. P. (1970). Theoretical foundations of principal factor analysis, canonical factor analysis, and alpha factor analysis. British Journal of Mathematical and Statistical

Psychology, 23, 1–21.

Neff, K. D. (2019). Setting the record straight about the Self-Compassion Scale. Mindfulness, 10(1), 200-202.

Neff, K. D. (2020). Commentary on Muris and Otgaar (2020): let the empirical evidence speak on the self-compassion scale. Mindfulness, 11, 1900-1909.

Rodriguez, A., Reise, S. P., & Haviland, M. G. (2016). Applying bifactor statistical indices in the

evaluation of psychological measures. Journal of Personality Assessment, 98, 223-237.

Reviewer #2: Overall, this paper does a good job of creating a Polish version of the SCS. Moreover, this paper (to the best of my knowledge) is the first to attempt to independently replicate Neff & Toth-Kiraly's findings with the SCS using bifactor ESEM analyses. For this reason, the paper makes an important contribution to the field. However, there are numerous problems with the write up of the paper that will need to be addressed before it can be published. Also, there are numerous grammatical errors in the paper, and the authors should have their paper carefully proof-read by a native English speaker before resubmission.

Introduction

The introduction to the paper focuses too much on the theoretical debate between whether or not the SCS should be used as two scores or one, and does not focus enough on psychometric issues. This is primarily a psychometric paper focused on validating the factor structure of the Polish SCS. The one versus two-factor debate should therefore be discussed primarily in terms of prior psychometric analyses using CFA that have found various factor solutions to the SCS. In particular, the authors need to discuss Neff's conceptualization of the components of self-compassion operating as a system and explain why she argues that bifactor ESEM is the best way to analyze multi-dimensional constructs that operate as a system.

Rather than stating that the 20-sample Neff, Toth-Kiraly et al (2019) paper "stresses the legitimacy of using a six-factor SCS structure or the overall score..." as if it was a theoretical paper, the authors should describe the empirical findings of that study, particularly because the current paper uses the same methods. They should describe how the single bifactor ESEM model of one general factor and six specific factors had the best fit, and also how the two-correlated bifactor ESEM model of two general factors with three specific factors each (positive and negative) had a good fit but inadequate factors loadings indicating that the positive and negative factors could not be differentiated. They should also stress that ESEM outperformed CFA for the SCS, and that this makes sense theoretically given that self-compassion is conceptualized as a multi-dimensional construct.

The long discussion of the link between self-compassion and mental health is not appropriate here and can be greatly reduced.

The authors state "The aim of the study was to adapt the Self-Compassion Scale by Kristin Neff on the basis of the methodological guidelines and instructions obtained from the test’s author." Instead, the authors should state that they are using the procedures laid out in Neff, Toth-Kiraly et al (2019) and highlight the importance of their analyses being replicated independently.

The authors should also give more general background on bifactor ESEM, either in the introduction or in the analyses section.

Study 1 Methods

The authors need to describe the SCS in their methods section, including how it is scored, sample items, and so on. The authors should indicate that negative items are reverse-coded so that they indicate the absence of self-judgment, isolation and overidentification.

Analyses. The authors should mention that ESEM uses target rotation (Browne, 2001), making it possible to rely on ESEM in a confirmatory manner as it allows the researcher to have more a priori control over the expected factor structure by targeting the cross-loadings to be as close to zero as possible, mimicking the specification of CFA. Many people mistakenly assume ESEM is like EFA and cannot be used in a confirmatory manner.

The authors need to state more clearly that they used bifactor analyses and explain what that method entails. They should also lay out why they test the particular models they do, including one general bifactor model versus two-correlated factor bifactor model, in line with Neff, Toth-Kiraly et al (2019).

The authors need to provide the goodness of fit cutoffs they are using.

Results

The authors should not repeat findings in the text that are given in the tables. The tables can speak for themselves and can be referred to without repeating their content.

Mean self-compassion scores in Table 6 should be calculated as they are by Neff in her recent papers - a mean is taken of each subscale and a grand mean is taken of the six subscales (after negative items are reverse-coded). Scores should be on a 5-point scale.

It is not clear when the authors reverse scored items and when they did not. It appears that they are not reverse scored in Table 7 but they are in Table 6? This needs to be clear as it's confusing now.

Study 2

The authors need to explain why they conducted a second study. Presumably it was to confirm the findings of Study 1 but they need to say so.

Methods. The authors should state that they used the same version of the SCS as used in Study 1.

Analyses. Why did the authors only use CFA and not ESEM? The authors should conduct the same analyses they conducted in Study 1 to see if they replicate in study 2. Also, given the excellent fit indices obtained I have a hard time believing that the authors used CFA and not ESEM. Please repeat all the analyses with study 2 as were done with study 1. Results can be put in the supplementary materials.

Once again, the authors should not give results in both the text and tables.

There is no reason to calculate correlations (Table 8) and regressions (Table 9). The authors should cut the regressions as they add no useful information over and above the correlations.

Discussion

The discussion should focus on the purpose of the paper, to create a valid Polish translation of the SCS. The authors should also highlight the fact that they independently replicated the findings of Neff, Toth-Kiraly et al. Finally, the authors should highlight the fact that their data confirms that a two-factor solution to the SCS is not valid whereas a single general factor and six specific factors is valid.

The authors appear to be confused about what constitutes psychometric validity for a scale. Strangely, they state: "Perhaps the division into self-compassion and self-coldness is more legitimate in the clinical groups. Many studies confirm the legitimacy of two-factor analysis, pointing to the significant role of self-coldness precisely among individuals experiencing disorders or difficulties in the psychological area (e.g. 17) , individuals experiencing other health problems (20) , or in the general population as a significant predictor of depressive symptoms (34)."

Of the studies they cite, only Costa (17) was actually a psychometric study. However, Costa used a two-factor CFA, an approach which the authors found was inadequate with their own data. Instead of saying it was legitimate because it was a clinical sample, the authors should instead point out the inadequacy of using CFA, and the superiority of the ESEM bifactor approach. Their findings clearly show that a two-factor solution is not psychometrically valid (using either CFA or bifactor ESEM), and clinical populations have nothing to do with it (the same results that the authors obtained were also obtained with clinical samples by Neff, Toth-Kiraly et al, 2019).

The studies which find that the negative subscales of the SCS (often termed coldness) predict depression and other negative outcomes more powerfully than the positive subscales (which can be termed warmth) does not "confirm the legitimacy of two-factor analysis." Instead, all these studies do is show that warmth and coldness predict outcomes differently. No sane person would say that because coldness predicts hand numbness more than warmth, that warmth and coldness must be measured separately. This is especially because warmth reduces coldness, just as compassionate self-responding reduces uncompassionate self-responding. They operate in tandem as a system, as clearly laid out by Neff, 2020. There is no known psychometric principle which states that the subscales of a multidimensional measure need to demonstrate the same strength of relationship with outcomes in order to justify use of a total score. The authors should rewrite this section or else drop it.

The authors should only present their data on the SCS and their other outcome variables such as personality or self-esteem as it relates to the validation of the Polish translation of the SCS, and not discuss findings independently of this context.

The authors state "Research by Charzyńska, Kocur, Działach, & Brenner (88) , carried out also on a Polish sample (but a different one), demonstrated the validity of analysing a two-factor model." The Charzynska paper did not demonstrate the validity of analyzing a two factor-model. It was not a psychometric paper. All that paper did was demonstrate that warmth and coldness have different associations with outcomes, which has nothing to do with the validity of a general factor versus two factors. If that study used the same translation of the Polish SCS used in this study, the authors instead need to clearly point out that the results of that study should be reconsidered because it used a two-factor solution to the SCS, which was shown to be psychometrically invalid in the current study.

6. PLOS authors have the option to publish the peer review history of their article (what does this mean?). If published, this will include your full peer review and any attached files.

Reviewer #1: No

Reviewer #2: No

---

## [Author Response · Author response to Decision Letter 0]

4 Apr 2022

Response to Reviewers’ Comments

Reviewer #1

Reviewer’s comment: Overall, I believe that this study has its merits as it tackles an important question. However, while reading the manuscript, I had some questions regarding the theoretical foundations and analyses conducted that might be helpful.

Our response: We appreciate the reviewer’s positive feedback on our study. We also thank the reviewer for the thoughtful comments and constructive suggestions, which have helped us improve the quality of the manuscript.

Reviewer’s comment: Because there are two alternative approaches to how self-compassion should be understood (Neff, with the notion that self-compassion is best represented by one global factor; and Gilbert, with the notion that self-compassion is best represented by two global factors), the authors should emphasise that their paper (and the SCS) follows Neff’s operationalisation of self-compassion.

Our response: We thank the reviewer for this comment. In Section 1.1, titled Tool design and factor analysis, we emphasise that the study is consistent with the theory developed by Kristin Neff.

Reviewer’s comment: The Introduction contains some repetition: the authors first discuss self-compassion and its correlates on page 3, but then bring up roughly the same idea on pages 8–9. These two parts could be merged to shorten the Introduction.

Our response: Thank you for your comment. We have shortened the excerpt on the first page, leaving the broader description only on pages 8–9.

Reviewer’s comment: While the authors did a good job of presenting competing views on self-compassion, I feel this could be done in a more balanced way. Currently, the two-factor representation has been discussed more. I would suggest that the authors incorporate the more recent studies by Neff (2019, 2020).

Our response: We appreciate the reviewer’s positive feedback on our study. We have added featured articles and expanded the introduction section. However, we believe that information on the positive and negative components of self-compassion should be included in the study, emphasising the current debate in the scholarly community. This debate is spurred on primarily by the numerous studies affirming the importance of separate self-compassion and self-coldness analyses (Muris & Petrocchi, 2017; Pfattheicher, Geiger, Hartung, Weiss, & Schindler, 2017; Lapez, Sanderman, & Schroevers, 2018) and those that affirm psychometric two-factor models (Brenner, Heath, Vogel, & Credé, 2017; Strickland, Nogueira-Arjona, Mackinnon, Wekerle, & Stewart, 2021).

Reviewer’s comment: A description of the SCS appears to be missing.

Our response: We have supplemented the description of the SCS.

Reviewer’s comment: It is unclear why the authors tested for gender differences. What is the importance of these tests? Is there a theoretical reason for conducting them?

Our response: We decided to analyse the gender differences because of previous inconclusive results (the scores of women have sometimes been reported as lower than those of men); this is possibly related to culture. A meta-analysis by Yarnell et al. (2015) showed that the differences between women and men were more significant in samples with a higher percentage of ethnic minorities. Perhaps ethnic minorities have more traditional gender roles than ethnic majority populations. In such a situation, women may emphasise meeting the needs of others over theirs. In Poland, we have a strongly patriarchal and more traditional culture. Hence, we assumed that the gender differences would be significant, which was confirmed.

Reviewer’s comment: While it seems that the authors followed the analytic plan outlined by Neff et al. (2019). They also do not seem to follow it closely. For example, there is no word on whether bifactor models (including those with one or two global factors), which is an important part of this study and the entire discussion on the most optimal representation of self-compassion.

Our response: In response to your suggestion, we have added three paragraphs to Subsection 1.1. Tool design and factor analysis, explaining the rationale behind higher-order and bifactor models and explaining why the bifactor model approach is suitable for conceptualisation of the self-compassion construct.

Reviewer’s comment: The various omega indices appear to be mixed up. Although it is true that all are derivatives of McDonald’s original work, the equation of the original omega index (let us call it composite reliability [CR] for the sake of simplicity) is different from the equation of the omega and omega hierarchical indices that have been developed specifically for bifactor models. This is because they refer to different aspects of reliability: CR refers to the actual reliability of a scale (akin to Cronbach’s alpha), while the other two refer to the amount of reliable variance attributed to the various factors. I suggest that the authors consult the research by McDonald (1970) and Rodriguez et al. (2016).

Our response: Indeed, we treated omega total as an index of composite reliability, presenting it together with Cronbach’s alpha. To avoid further confusion, we decided not to present this coefficient in Table 5, as it is primarily an additional coefficient for estimating composite reliability—therefore, it may be considered redundant. 

Reviewer’s comment: When describing the alternative factor solutions, the authors spend way too much time on the improper solutions. Given the length of the paper, I would suggest simply reporting all fit indices in a table and then quickly moving on to the comparison of the most adequate solutions. Again, consult Neff et al. (2019) on the structure for reporting on these models concisely and which aspects of the models (factor loadings, cross-loadings, or factor correlations) to report. That is to say, the authors should first focus on the comparison of the first-order solutions and then retain one. Regarding the bifactor models, first test whether one or two global factors should be incorporated (focus on the definition of the global factors and the correlations between the two global factors), retain one of the bifactor models and compare this bifactor model to its first-order counterpart.

Our response: We have simplified the report using the sequence you proposed. First, we analysed all indices with a good fit, both for the first-order solution and the bifactor solutions. We then retained one first-order solution and two-bifactor solutions. 

Reviewer’s comment: According to APA guidelines, it is redundant to present detailed results both in tables/figures and in the text.

Our response: We have omitted repetitive information on results in the text that was also presented in a table. 

Reviewer’s comment: Factor loading tables: add significance values to all factor loadings instead of using italics; there is plenty of space left.

Our response: In the case of the tables with information on factor loadings, we reported the results in a similar manner as Neff et al. (2018). Furthermore, bearing in mind the volume of information presented in these tables, we are convinced that additional information will make the tables less clear. However, if the reviewer believes that this change is essential to the quality of the study, we can provide such information as supplementary material. 

Reviewer’s comment: The degrees of freedoms of the single-bifactor model (it is unclear whether it is a CFA-based or ESEM-based model) reported in Study II do not match the degrees of freedoms in Study I, which means that different models were estimated. This has me deeply concerned about the results.

Our response: Thank you for noticing this. Indeed, we mistakenly presented a model in Study II that was estimated differently from the model in Study I. This was because we used a different software, as was noted in the opening description of Study II. This has been corrected, and consistent with Study I, the new set of results was derived using the Mplus software. The computations for fit indices have changed, but the interpretation of the results remains the same. After corrections, the description of the results is stated as follows: ‘A close examination of the fit indices reveals that the six-factor model has an unsatisfactory fit in CFA: χ2 (284) = 1970.55, p < 0.001, CFI = 0.89, TLI = 0.88, RMSEA = 0.09 [90% CI: 0.089–0.097]. However, the six-factor model achieved a satisfactory fit using ESEM: χ2 (184) = 561.55, p < 0.001, CFI = 0.98, TLI = 0.96, RMSEA = 0.06 [90% CI: 0.049–0.060]. Using the ESEM procedure, the single-bifactor model yielded a satisfactory fit: χ2 (164) = 442.07, p < 0.001, CFI = 0.99, TLI = 0.97, RMSEA = 0.04 [90% CI: 0.044–0.055], which is superior to that achieved using CFA: χ2 (273) = 2626.21, p < 0.001, CFI = 0.85, TLI = 0.82, RMSEA = 0.11 [90% CI: 0.108–0.116]. Furthermore, comparing the fit of the bifactor and six-factor models achieved using the ESEM procedure, the bifactor model achieved a better fit. Therefore, we conclude that the single-bifactor solution provides a slightly better representation of the data than the six-factor model does.’

Reviewer’s comment: Where is the limitations section?

Our response: We have supplemented the limitations section.

Reviewer’s comment: There are a few sections in the paper which are not presented in their correct location. For example, the authors should not discuss prior ESEM-based studies and results in the analyses section. Similarly, the authors should not talk about the ethical aspects of their study in the Present Study section.

Our response: Thank you for this suggestion. Therefore, we have added another subsection to the Introduction, i.e., Subsection 1.2. Exploratory structural equation modelling (ESEM) in testing the theoretical structure of self-compassion. In this subsection, we discuss ESEM-based research. Furthermore, we have moved all the ethical aspects of the research from the ‘Present study’ subsection into a footnote. 

Reviewer’s comment: Finally, there are a fair few typos and unusual sentence structures throughout the manuscript, which made it difficult to understand parts of the manuscript. Thus, I would suggest that the authors ask a colleague (whose native language is English) to read the paper before submission.

Our response: We appreciate the reviewer’s careful attention to our use of English in the manuscript. We have put our manuscript through repeated proofreading by a professional editing service.

Reviewer #2

Reviewer’s comment: The introduction to the paper focuses too much on the theoretical debate between whether or not the SCS should be used as two scores or one score and does not focus enough on psychometric issues. This is primarily a psychometric paper focused on validating the factor structure of the Polish SCS. The one-factor versus two-factor debate should therefore be discussed primarily in terms of prior psychometric analyses using CFA that have found various factor solutions to the SCS. In particular, the authors need to discuss Neff’s conceptualisation of the components of self-compassion operating as a system and explain why she argues that bifactor ESEM is the best way to analyse multidimensional constructs that operate as a system.

Our response: Heeding this suggestion, we have added three paragraphs to Subsection 1.1. Tool design and factor analysis, explaining the rationale behind higher-order models and bifactor models and why the bifactor model approach is suitable for conceptualisation of the self-compassion construct. Furthermore, we more broadly discussed the rationale for applying the ESEM approach to self-compassion conceptualisations in Subsection 1.2.

Reviewer’s comment: Rather than stating that the 20-sample Neff, Toth-Kiraly et al. (2019) paper ‘stresses the legitimacy of using a six-factor SCS structure or the overall score...’ as if it were a theoretical paper, the authors should describe the empirical findings of that study, particularly because this study uses the same methods. They should describe how the single-bifactor ESEM model of one general factor and six specific factors had the best fit and how the correlated two-bifactor ESEM model of two general factors with three specific factors each (positive and negative) had a good fit but inadequate factor loadings, indicating that the positive and negative factors could not be differentiated. They should also stress that ESEM outperformed CFA for the SCS and that this makes sense theoretically given that self-compassion is conceptualised as a multidimensional construct.

Our response: We have described the empirical findings of the Neff et al. (2019) study in more specific detail in the specified paragraph.

Reviewer’s comment: The long discussion of the link between self-compassion and mental health is not appropriate here and can be greatly reduced.

Our response: We thank the reviewer for this comment. We have shortened the section Self-compassion vs. mental health and well-being and retained only the most essential information.

Reviewer’s comment: The authors state, ‘The aim of the study was to adapt the Self-Compassion Scale by Kristin Neff on the basis of the methodological guidelines and instructions obtained from the test’s author.’ Instead, the authors should state that they are using the procedures laid out in Neff, Toth-Kiraly et al. (2019) and highlight the importance of their analyses being replicated independently.

Our response: Thank you very much for this suggestion. We have applied your recommendation at the end of the first paragraph of Section 1.2. as follows: ‘Furthermore, we decided to use the procedures proposed by Neff et al., [42] bearing in mind the importance of replicating their results using slightly different samples. Most of the samples assessed in the Neff et al. research can be considered as either western, educated, industrialised, prosperous or democratic, with a few exceptions (e.g., Iran). Therefore, it is essential to present additional evidence of the validity of those results by examining the possibility of their extrapolation into less typical samples (e.g., Poland).’

Reviewer’s comment: The authors should also provide a more general background on bifactor ESEM, either in the introduction or in the analyses section.

Our response: To effect this recommendation, we have added an additional paragraph to Subsection 1.2. Exploratory structural equation modelling (ESEM) in testing the theoretical structure of self-compassion, as follows: ‘Furthermore, some research findings indicate that a bifactor model provides a superior theoretical conceptualisation of self-compassion than that yielded by various other higher-order models. This is because various behaviours related to the content of individual scale items may be considered representations of the general level of self-compassion and its specific components.[27,42] By these terms, the ESEM procedure appears to be the evidently more appropriate analytical approach. Most importantly, it permits estimation of both the general and the unique relationship between scale items (i.e., establishing the relationships between specific item groups and both the general self-compassion factor and its specific subcomponents) and encourages examination of the highly sophisticated interaction within the system between items and their cross-loadings. In such circumstances, the general factor(s) in bifactor ESEM models are typically specified similarly as in CFA models, i.e., assuming no cross-loadings between the factors. However, the specific factors are specified as ESEM factors, i.e., the cross-loadings between specific factors are allowed to vary and are not necessarily equal to zero.[16]’

Reviewer’s comment: The authors need to describe the SCS in their methods section, including how it is scored, sample items, and so on. The authors should indicate that negative items are reverse coded so that they indicate the absence of self-judgement, isolation and overidentification.

Our response: We are thankful to the reviewer for this particularly useful observation. In the revised manuscript, we have supplemented the description of the SCS with the inclusion of exemplary items for each of the six subscales. We also emphasised that the items on negative scales are reverse coded. This will undoubtedly make the manuscript more comprehensive.

Reviewer’s comment: The authors should mention that ESEM uses target rotation (Browne, 2001), making it possible to rely on ESEM in a confirmatory manner, as it allows the researcher to have more a priori control over the expected factor structure by targeting the cross-loadings to as close to zero as possible, mimicking the specifications of CFA. Many people mistakenly assume that ESEM is like EFA and cannot be used in a confirmatory manner.

Our response: To implement the recommendation in this comment, we have added an additional paragraph to Subsection 1.2., as follows: ‘Notably, the ESEM procedure covers target rotation, which allows the researcher to have a priori control over the hypothesised factor structure by assuming the cross-loadings to be, if possible, close to zero—but not equal to zero, as in CFA. Therefore, it is practical to rely on ESEM in a confirmatory manner (in contrast to EFA, which is typically used to explore the potential theoretical structures of a construct), i.e., as a way of validating the theoretical structure of a given construct.’

Reviewer’s comment: The authors need to state more clearly that they used bifactor analyses and explain what that method entails. They should also lay out why they test the particular models they do, including one general bifactor model versus the two-correlated factor bifactor model, in line with Neff, Toth-Kiraly et al. (2019).

Our response: To implement this suggestion, we have added three paragraphs to Subsection 1.1. Tool design and factor analysis, explaining the rationale behind the higher-order model and the bifactor model and explaining why the bifactor model approach is suitable for conceptualisation of the self-compassion construct.

Reviewer’s comment: The authors need to provide the goodness-of-fit cut-offs they are using.

Our response: To meet this requirement, we added a paragraph to Section 2.1.3. Analyses, which reads as follows: ‘All factor loadings were fully standardised, and for indicators of adequacy of fit, we used a comparative fit index (CFI) and the Tucker–Lewis index (TLI), >0.90, the root mean square error of approximation (RMSEA), <0.08 (with its 90% confidence intervals), and the weighted root mean square residual (WRMR), <1.0. We also report the chi-square values, together with their statistical significance.’

Reviewer’s comment: The authors should not repeat findings in the text that are given in the tables. The tables can speak for themselves and can be referred to without repeating their content.

Our response: To eliminate such repetition, we decided to omit information on the results in the text that are also present in a table.

Reviewer’s comment: The mean self-compassion scores in Table 6 should be calculated as calculated by Neff in her recent papers: a mean is taken of each subscale and a grand mean is taken of the six subscales (after negative items are reverse coded). Scores should be on a 5-point scale.

Our response: We have recalculated all means using the method outlined in the comment.

Reviewer’s comment: It is not clear when the authors reverse scored items and when they did not. It appears that they did not reverse score the items in Table 7 but did in Table 6. This needs to be clear, as it is currently confusing.

Our response: We have added additional information on reverse coding under Table 6, as follows: ‘General self-compassion scores were calculated by reverse coding the self-judgement, isolation and overidentification items, then summing all six subscale means. For the sake of simplicity interpreting the subscales, scores on self-judgement, isolation and overidentification were calculated based on non-reverse-coded answers.’ 

Reviewer’s comment: The authors need to explain why they conducted a second study. Presumably, it was to confirm the findings of Study I, but they need to say so.

Our response: Thank you for this suggestion. To address this, we included an additional statement at the end of Section 1.3, which reads as follows: ‘Subsequently, the psychometric properties of the tool were analysed in two successive studies: Study I, and Study II. We decided to conduct two independent studies to increase the validity of our results and formulate highly robust conclusions regarding the stability of the theoretical structure of self-compassion among Polish samples.’

Reviewer’s comment: The authors should state that they used the same version of the SCS used in Study I.

Our response: We thank you for this observation. We have modified the description in Study II to include this information.

Reviewer’s comment: Why did the authors only use CFA and not ESEM? The authors should conduct the same analyses in Study II that they conducted in Study I to see if they replicate the results. Also, given the excellent fit indices obtained, I have a hard time believing that the authors used CFA and not ESEM. Please repeat all the analyses in Study II that were performed in Study I. The results can be presented as supplementary material.

Our response: Thank you for noticing this. Indeed, in Study II, we mistakenly presented a model that was estimated differently from the model in Study I. This was because we used a different software, which was noted in the opening description of Study II. This has been corrected, and consistent with Study I, the new set of results was derived using the Mplus software. The computations for fit indices have changed, but the interpretation of the results remains the same. After corrections, the description of the results is stated as follows: ‘A close examination of the fit indices reveals that the six-factor model has an unsatisfactory fit in CFA: χ2 (284) = 1970.55, p < 0.001, CFI = 0.89, TLI = 0.88, RMSEA = 0.09 [90% CI: 0.089–0.097]. However, the six-factor model achieved a satisfactory fit using ESEM: χ2 (184) = 561.55, p < 0.001, CFI = 0.98, TLI = 0.96, RMSEA = 0.06 [90% CI: 0.049–0.060]. Using the ESEM procedure, the single-bifactor model yielded a satisfactory fit: χ2 (164) = 442.07, p < 0.001, CFI = 0.99, TLI = 0.97, RMSEA = 0.04 [90% CI: 0.044–0.055], which is superior to that achieved using CFA: χ2 (273) = 2626.21, p < 0.001, CFI = 0.85, TLI = 0.82, RMSEA = 0.11 [90% CI: 0.108–0.116]. Furthermore, comparing the fit of the bifactor and six-factor models achieved using the ESEM procedure, the bifactor model achieved a better fit. Therefore, we conclude that the single-bifactor solution provides a slightly better representation of the data than the six-factor model does.’

Reviewer’s comment: Once again, the authors should not give results in both the text and tables.

Our response: Based on previous similar comments, we decided to omit the repetitive information on results in text that was also present in a table.

Reviewer’s comment: There is no reason to calculate correlations (Table 8) and regressions (Table 9). The authors should cut out the regressions, as they add no useful information over and above the correlations.

Our response: With respect to this feedback from the reviewer, we decided to omit that section.

Reviewer’s comment: The discussion should focus on the purpose of the paper: to create a valid Polish translation of the SCS. The authors should also highlight the fact that they independently replicated the findings of Neff, Toth-Kiraly et al. Finally, the authors should highlight the fact that their data confirms that a two-factor solution to the SCS is not valid, whereas a single general factor and six specific factors is valid.

Our response: Thank you for your suggestion. We supplemented the discussion section and developed the limitations and conclusion section. We have also highlighted that our study replicates the findings of Neff, Toth-Kiraly et al. (2019) and described in detail how our data confirms a two-factor solution.

Reviewer’s comment: The authors appear to be confused about what constitutes psychometric validity for a scale. Strangely, they state, ‘Perhaps the division into self-compassion and self-coldness is more legitimate in the clinical groups. Many studies confirm the legitimacy of two-factor analysis, pointing to the significant role of self-coldness precisely among individuals experiencing disorders or difficulties in the psychological area (e.g., 17), individuals experiencing other health problems (20), or the general population as a significant predictor of depressive symptoms (34).’

Our response: To avoid confusion, we moved this paragraph to the limitations section, where we described our perspective of the two-factor structure of the SCS. 

Reviewer’s comment: Of the studies they cite, only Costa (17) was actually a psychometric study. However, Costa used a two-factor CFA, an approach which the authors found to be inadequate with their own data. Instead of saying it was legitimate because it was a clinical sample, the authors should point out the inadequacy of using CFA and the superiority of the ESEM bifactor approach. Their findings clearly show that a two-factor solution is not psychometrically valid (using either CFA or bifactor ESEM) and clinical populations have nothing to do with this (the same results obtained by the authors were also obtained by Neff, Toth-Kiraly et al., 2019 using clinical samples).

Our response: Thank you for this suggestion. We have rephrased this part of the manuscript to provide a more comprehensive discussion of that topic. It reads, ‘Furthermore, our findings highlight some limitations of CFA, which is often used as a routine procedure while testing the theoretical structures of psychological constructs. The research conclusions cited in this study may have resulted from theoretical inadequacies and simplifications generated while using CFA, e.g., fixing cross-loadings as zero. In this regard, our findings demonstrate the superiority of the ESEM bifactor approach over CFA. However, psychometric studies that employ ESEM are still much more unique examples than typical procedures. Hence, we recommend expanding the number of studies that use ESEM procedures in general and conducting more research on self-compassion in clinical samples. Such endeavours should be worthwhile, as they might produce an unambiguous solution to the debate.’

Reviewer’s comment: The studies which find that the negative subscales of the SCS (often termed coldness) predict depression and other negative outcomes more powerfully than the positive subscales (which can be termed warmth) do not ‘confirm the legitimacy of two-factor analysis.’ Instead, all that these studies do is show that warmth and coldness predict outcomes differently. No sane person would say that because coldness predicts hand numbness more than warmth, then warmth and coldness must be measured separately. This is especially because warmth reduces coldness, just as compassionate self-responding reduces uncompassionate self-responding. They operate in tandem as a system, as clearly laid out by Neff (2020). There is no known psychometric principle which states that the subscales of a multidimensional measure need to demonstrate the same strength of relationship with outcomes to justify use of a total score. The authors should rewrite this section or omit it.

Our response: Thank you for this suggestion. However, our review of the literature shows that the distinction between self-compassion and self-coldness cannot be ignored completely (Costa et al., 2016; Brenner et al., 2018; Muris & Otgaar, 2020; Strickland, Nogueira-Arjona, Mackinnon, Wekerle, & Stewart, 2021;). First, some psychometric studies have indicated the legitimacy of the division into self-compassion and self-coldness. For example, a study by Brenner, Heath, Vogel, & Credé (2017) shows that the bifactor model comprising two general factors (self-compassion and self-coldness) and six specific factors demonstrates the best fit to the data. Additionally, there is the recent study by Strickland, Nogueira-Arjona, Mackinnon, Wekerle, & Stewart (2021), with results that strongly support a two-factor hierarchical model with six lower-order factors. A general self-compassion factor was not supported at the higher-order or lower-order levels.

Second, the analogy of warming hands is interesting, but when we look closely at the positive and negative components, we see that they are not one-dimensional—like warm and cold hands. For example, an internal critic can coexist with self-kindness, just as increasing self-kindness does not always diminish internal criticism. A person may have immense love for themself and simultaneously hold themselves to high expectations, which is associated with a critical evaluation of their achievements in, for example, educational or professional spheres. In summary, our research validates one-way analysis, which we recommend in the manuscript. However, we believe it would be unjustifiable to ignore the many studies and psychometric factor analysis data that support the two-factor model.

Reviewer’s comment: The authors should only present their data on the SCS and their other outcome variables, such as personality or self-esteem, as it relates to the validation of the Polish translation of the SCS, and not discuss findings outside of this context.

Our response: In response to the reviewer’s concerns, we decided to shorten that part of the discussion.

Reviewer’s comment: The authors state, ‘Research by Charzyńska, Kocur, Działach, & Brenner (88), carried out also on a Polish sample (but a different one), demonstrated the validity of analysing a two-factor model.’ The Charzyńska paper did not demonstrate the validity of analysing a two-factor model. It was not a psychometric paper. All that paper did was demonstrate that warmth and coldness have different associations with outcomes, which has nothing to do with the validity of a general factor versus two factors. If that study used the same translation of the Polish SCS used in this study, the authors instead need to clearly point out that the results of that study should be reconsidered because it used a two-factor solution to the SCS, which was shown to be psychometrically invalid in the current study.

Our response: Thank you for noticing this. We were informed via private correspondence that in the study titled Testing the indirect effect of type 1 diabetes on life satisfaction through self-compassion and self-coldness, the SCS structure was examined before making the calculations (E. Charzyńska, personal communication, June 25, 2020). The first author sent us the calculations and their descriptions via e-mail. Using confirmatory factor analysis, her team tested several models of the SCS structure: (1) a single factor (i.e., unidimensional) model; (2) an oblique two-factor model; (3) an oblique six-factor model; (4) a higher-order one-factor model; (5) a higher-order two-factor model; (6) a bifactor model; and (7) a two-tier bifactor model. The results show that only two models demonstrated an acceptable model fit: the two-tier bifactor model and the higher-order two-factor model. The results of a χ2 difference test indicate that the former model is superior to the latter; thus, they retained a two-tier bifactor model as the best fitting model. 

Nevertheless, in our revised manuscript, we have removed the reference to the Charzyńska et al. (2020) study to avoid causing confusion.

Furthermore, in the limitations section, we have added a detailed discussion in which we communicate that despite our study validating the single-bifactor conceptualisation of self-compassion and our recommending using the SCS in that manner, we also believe that the division into compassion and self-coldness is also critical. Indeed, many studies confirm the legitimacy of two-factor analysis, pointing to the significant role of self-coldness (Körner et al., 2015; Muris & Petrocchi, 2017; Muris & Otgaar, 2020). Other studies also point to the psychometric legitimacy of analysing a two-factor model, taking into account self-compassion and self-coldness (López et al., 2015; Costa et al., 2016; Brenner et al., 2018; Strickland et al., 2021). 

The results quoted and the contradictions they contain show that the debate on the factor structure of SCS is highly complex and multifaceted, and we are convinced that this debate should continue.

---

## [Editor Report · Decision Letter 1]

13 Apr 2022

Validity and reliability of the Polish version of the Self-Compassion Scale and its correlates

PONE-D-21-04597R1

Dear Dr. Kocur,

We’re pleased to inform you that your manuscript has been judged scientifically suitable for publication and will be formally accepted for publication once it meets all outstanding technical requirements.

Kind regards,

Frantisek Sudzina

Academic Editor

PLOS ONE
---

## [Editor Report · Acceptance letter]

18 Apr 2022

PONE-D-21-04597R1 

Validity and reliability of the Polish version of the Self-Compassion Scale and its correlates 

Dear Dr. Kocur:

I'm pleased to inform you that your manuscript has been deemed suitable for publication in PLOS ONE. Congratulations! Your manuscript is now with our production department. 

Kind regards, 

on behalf of

Dr. Frantisek Sudzina 

Academic Editor

PLOS ONE